# Immersive NREM2 dreaming preserves subjective sleep depth against declining sleep pressure

Adriana Michalak[1,2☉], Davide Marzoli[1☉], Francesco Pietrogiacomi[1☉], Damiana Bergamo[1,3], Valentina Elce[1], Bianca Pedreschi[1], Giorgia Mosca[1], Alessandro Navari[4], Michele Emdin[4,5], Emiliano Ricciardi[1], Giacomo Handjaras[1], Giulio Bernardi[1]*

1 MoMiLab Research Unit, IMT School for Advanced Studies Lucca, Lucca, Italy, 2 Department of Biomedical, Metabolic, and Neural Sciences, University of Modena and Reggio-Emilia, Modena, Italy, 3 Department of General Psychology, University of Padova, Padova, Italy, 4 Cardiovascular Medicine Department, Fondazione Toscana Gabriele Monasterio, Pisa, Italy, 5 Health Science Interdisciplinary Center, Scuola Superiore Sant'Anna, Pisa, Italy

☉ These authors contributed equally to this work.
* giulio.bernardi@imtlucca.it

## Abstract

Perceived sleep depth is a key determinant of subjective sleep quality, traditionally thought to reflect unconsciousness and reduced cortical activation. Here, we combined high-density EEG with a serial awakening paradigm during NREM2 (N2) sleep in healthy human participants to examine its neural and experiential correlates. As expected, deeper sleep was associated with reduced cortical activation, reflected in a lower high-to-low frequency power ratio. Yet, this relationship weakened in the presence of dreaming, indicating that immersive conscious experiences may counteract the impact of cortical activation on perceived depth. Indeed, perceived sleep depth was lowest during minimal forms of awareness characterized by a mere sense of presence, and highest during immersive dreaming or deep unconsciousness. Across the night, physiological sleep pressure and subjective sleepiness declined, but perceived sleep depth rose alongside increasing dream immersiveness. These findings challenge the view that the feeling of deep sleep arises solely from reduced brain activity and suggest instead that immersive dreaming may help sustain the subjective experience of deep sleep as homeostatic pressure wanes.

## Introduction

The key to a good night's sleep lies not only in its duration, but also in the subjective impression of having slept deeply and without interruption. Yet the neural underpinnings of this perception remain poorly understood.

The transition from wakefulness to sleep is marked by a shift in brain activity from fast, desynchronized EEG patterns to slower, more synchronized rhythms characteristic of nonrapid-eye-movement (NREM) sleep. Conventional sleep scoring defines a progression through NREM stages N1 to N3, characterized by an increasing

**Data availability statement:** The deidentified data generated in this study and the custom-made MATLAB codes used for data analysis and to reproduce main and supplementary figures and tables presented in this work have been deposited in the Zenodo database (DOI: https://doi.org/10.5281/zenodo.15372992).

**Funding:** This work was supported by an ERC Starting Grant (#948891; to G.B.), by the "Resilienza Economica e Digitale" project (CUP D67G23000060001) funded by the Italian Ministry of University and Research (MUR) as "Department of Excellence" (Dipartimenti di Eccellenza 2023-2027, Ministerial Decree no. 230/2022), and by the "Tuscany Health Ecosystem—THE" Project, Spoke 8, granted by Next Generation EU—National Recovery and Resilience Plan (Piano Nazionale di Ripresa e Resilienza, NRRP)—Mission 4 Component 2 Investment 1.4—Ministry of University and Research (MUR) Call N. 3277 (Project Code ECS_00000017; to E.R., G.B., and G.H.). G.B. is also supported by a FARE2020 grant (#R2049C8YN3) and a PRIN grant (#2022BNE97C) of the Italian Ministry of University and Research (MUR). The funders had no role in the conceptualization, design, data collection, analysis, decision to publish, or preparation of the manuscript.

**Competing interests:** The authors have declared that no competing interests exist.

**Abbreviations:** AASM, American Academy of Sleep Medicine; AIC, akaike information criterion; BIC, bayesian information criterion; CE, conscious experience; CESP, conscious experience with a sense of presence; CEWR, conscious experience without recall of content; CI, confidence intervals; ECG, electrocardiography; EMG, electromyography; EOG, electrooculography; FDR, false discovery rate; GLME, generalized linear mixed-effects; ICA, independent component analysis; IQR, interquartile range; LRT, likelihood ratio test; N2, NREM2; NCE, no conscious experience; NREM, non-rapid-eye-movement; PCA, principal component analysis; PC1, "perceptual immersion" principal component; PC2, "reflective thought" principal component; PSD, power spectral density; rCEWR, rich conscious experience without recall of content; sCEWR, simple conscious experience without recall of content; UNC, unconsciousness.

occurrence of large-amplitude, low-frequency slow waves (<4 Hz), reflecting synchronized alternations between neuronal firing and silence, and by a concomitant reduction in high-frequency (>25 Hz) activity [1]. This physiological progression is paralleled by increased arousal thresholds in response to external stimuli [2] as well as a decline in the frequency and narrative complexity of reported conscious experiences [3]. In line with this, recent high-density EEG investigations showed that NREM dream experiences emerge from localized shifts towards wake-like activity patterns marked by reduced slow-wave activity and increased high-frequency power [4]. These observations have led to the widely held assumption that sleep slow waves may be the primary drivers of unconsciousness, sensory disconnection, and subjective sleep depth [5,6]. However, REM sleep, a stage characterized by wake-like cortical activity and rich, immersive conscious experiences, also has a high arousal threshold and is subjectively perceived as deep [7]. Moreover, serial awakening studies revealed marked fluctuations in consciousness level and subjective sleep depth even within the same sleep stage [8]. Together, these considerations point to a more complex relationship between cortical activation patterns, perceived sleep depth, and ongoing mental activity than is commonly assumed.

To investigate this complexity, we analyzed a large dataset of 196 overnight high-density EEG recordings (256 electrodes) from 44 healthy adults (mean age = 26.4 ± 3.9 years; 21 female, 23 male), collected using a serial awakening paradigm specifically targeting NREM2 (N2) sleep. This stage was chosen because it represents the largest proportion of total sleep (~50%), is distributed throughout the whole sleep period, and exhibits broad variability in both the presence of dream experiences and perceived sleep depth [9]. By focusing on a single sleep stage, we also aimed to avoid fragmenting other sleep stages and thereby altering their associated physiological functions. The dataset combined two experiments based on similar protocols. In both experiments, each participant completed four nights of sleep in the lab, during which they were awakened repeatedly to report on prior mental activity and to rate both their perceived sleep depth and their subjective sleepiness. This procedure yielded a total of 1,024 awakenings with corresponding subjective reports (Figs 1, S1, S2; S1 Table).

Building on this dataset, we sought to identify the neural and experiential correlates of subjective sleep depth during N2. Specifically, we examined whether variations in conscious experience, ranging from complete unconsciousness to immersive dreaming, modulate the relationship between cortical activation and the feeling of deep sleep. Our results reveal that subjective sleep depth varies systematically with both neural activity and dream phenomenology, indicating that immersive conscious experiences may help sustain the subjective sense of deep sleep even as physiological sleep pressure declines.

## Results

### Conscious experience alters the mapping between brain activity and perceived sleep depth

We first examined the neural correlates of perceived sleep depth during N2 sleep and of subjective sleepiness reported upon awakening (Figs 2 and S3). Specifically,

**A   Report type distribution**

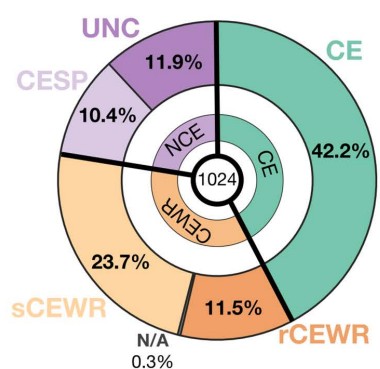

**B   Distribution of report types throughout the night**

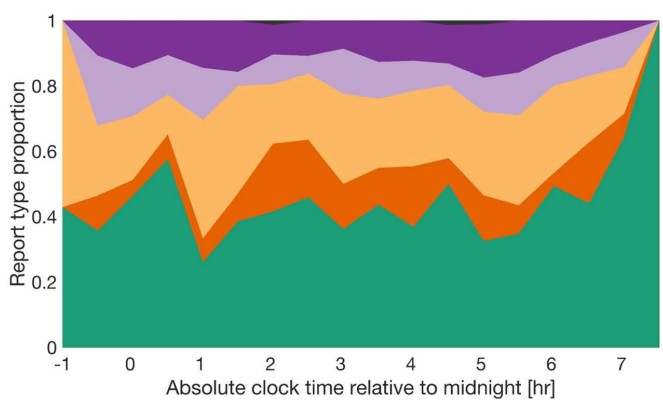

**Fig 1. Dataset description. (A)** Distribution of report types across the full dataset according to the classical three-level classification of reports (CE, CEWR, and NCE) and to the five-level classification explored in the present study (CE, rCEWR, sCEWR, CESP, and UNC). CE (green): conscious experience with recall of content; CEWR (orange): conscious experience without recall of content, further divided into rich (rCEWR, dark orange) and simple (sCEWR, light orange); NCE (purple): no reported conscious experience, further divided into conscious experience with a sense of presence (CESP, light purple) and true unconsciousness (UNC, dark purple). In the CEWR group, information about the rich/simple subtype was unavailable for a few reports (0.29%) due to misclassification at the time of acquisition. The dataset included a total of 1,024 awakenings from N2 sleep. **(B)** Proportion of report types as a function of time of night (30-min bins). The x-axis represents time relative to midnight (from –1 to +7.5 hours). Only five reports occurred outside this interval. The data and code used to generate this figure are openly available (https://doi.org/10.5281/zenodo.15372992).

EEG activity in the two minutes preceding each awakening was analyzed to obtain three frequency-based metrics: low-frequency power (*delta*: 0.5–4 Hz), high-frequency power (*gamma*: 25–50 Hz), and the high-to-low-frequency power ratio (*gamma/delta*), a marker of overall cortical activation [4,8,10]. These frequency bands were selected based on prior work linking them to both dreaming and subjective sleep depth. Dreaming has been associated with reduced *delta*, increased *gamma*, and a higher *gamma/delta* ratio, reflecting localized cortical activation during sleep [4]. Likewise, subjective sleep depth has been shown to vary with the balance between low- and high-frequency cortical activity [8,11].

Analyses were conducted using Generalized Linear Mixed-Effects (GLME) models at the electrode level, with cluster-mass correction applied across electrodes (cluster-forming threshold: *uncorrected p* < 0.005, minimal cluster size: 3 electrodes, number of permutations: 5,000; see Materials and methods). Each model included *experiment*, *night*, and *time* of night (relative to midnight) as fixed effects and *participant* as a random intercept to account for interindividual variability. For clarity, only significant results are reported in the main text, while full model outputs are provided in the Supporting information.

These analyses revealed a widespread negative association between subjective sleep depth and both *gamma* power ($p < 0.05$; range across electrodes, $\beta = [-0.505, -0.190]$) and the high-to-low-frequency power *ratio* ($\beta = [-0.258, -0.115]$; Fig 2A; S2 Table). A similar pattern was observed for subjective sleepiness upon awakening, with significant negative associations for *gamma* power ($\beta = [-0.315, -0.148]$) and the power *ratio* ($\beta = [-0.135, -0.091]$); Fig 2C; S3 Table). In contrast, *delta* power was not significantly associated with either subjective sleep depth or sleepiness, although the direction of the effect was mostly positive for both measures.

To assess whether these brain–behavior relationships were influenced by the presence of conscious experiences during sleep, we next categorized awakenings into three experience types based on retrospective reports collected immediately after awakening [4,12]: '*conscious experiences*' (CE, 432 reports; participant-wise mean and standard deviation: 41.15 ± 21.47%) when participants recalled a subjective experience, '*conscious experiences without recall of content*'

## Relationship between brain activity indices and subjective sleep depth

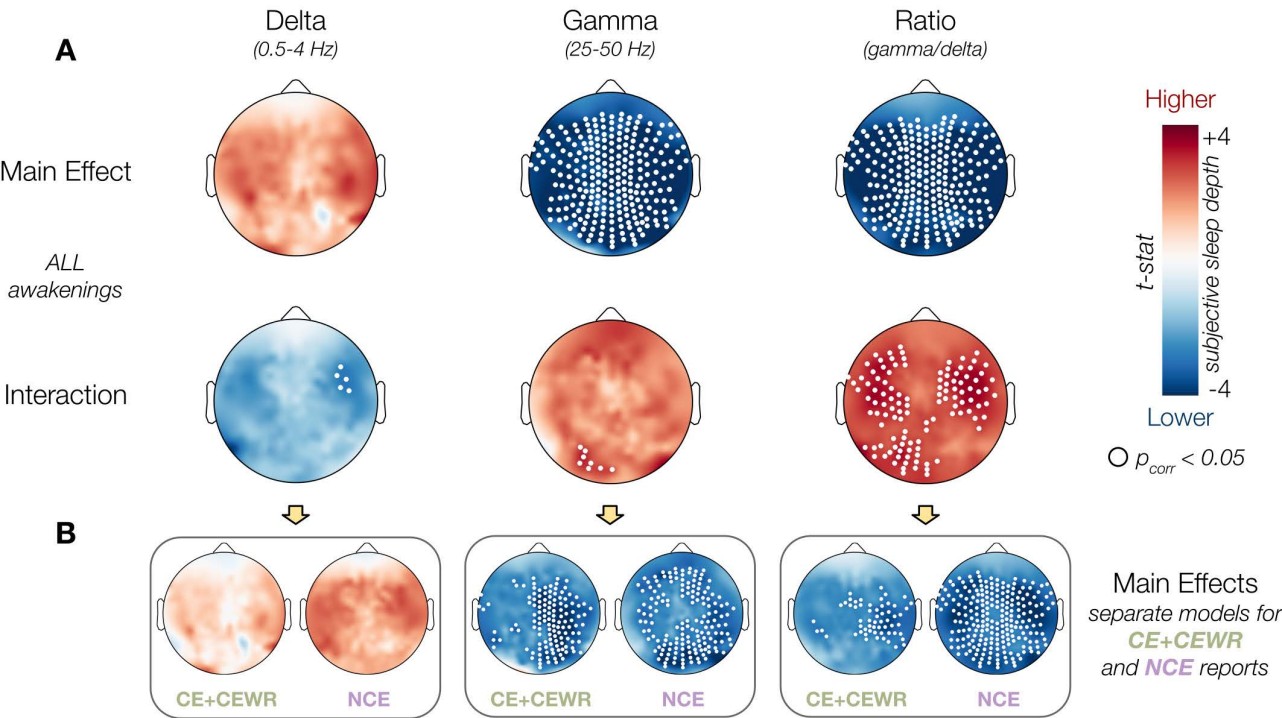

## Relationship between brain activity indices and subjective sleepiness

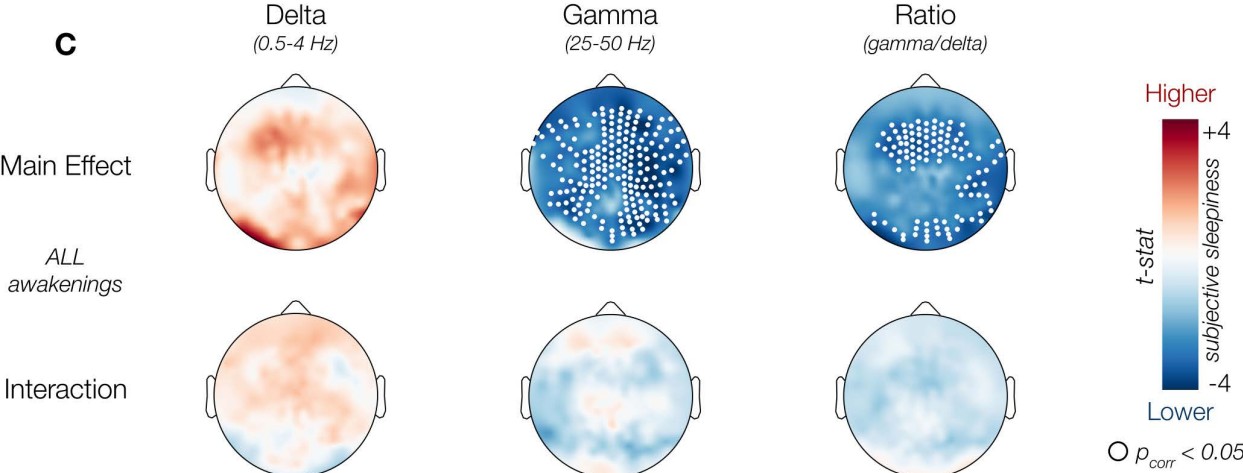

**Fig 2. Neural correlates of subjective sleep depth and subjective sleepiness. (A, C)** Scalp topographies illustrate the results of GLME models assessing the relationship between EEG activity (delta power, gamma power, and gamma/delta ratio) and subjective sleep depth (A) or subjective sleepiness (C). The top rows show the main effects of each brain activity index; the bottom rows show the interaction between brain activity and conscious experience (CE+CEWR vs. NCE). Significant interaction effects were found for subjective sleep depth but not sleepiness. **(B)** Follow-up models for sleep depth showing separate effects for conscious (CE+CEWR) and unconscious (NCE) reports. All GLME models included participant as a random intercept and experiment, night, and time of night (relative to midnight) as fixed effects. Each index was tested in a separate model. Wald statistics are projected on the scalp for each electrode; white dots indicate significant clusters after cluster-mass correction (corrected $p < 0.05$). The data and code used to generate this figure are openly available (https://doi.org/10.5281/zenodo.15372992).

(CEWR, 364 reports; 35.60 ± 20.12%) when participants reported the impression that they had an experience but could not recall its content (so-called '*white dreams*'), and '*no conscious experience*' (NCE, 228 reports; 23.25 ± 18.37%) when no mental activity was reported (Fig 1).

We then tested for interactions between brain activity and the presence (CE+CEWR) or absence (NCE) of conscious experience. This analysis revealed that the relationship between EEG features and subjective sleep depth was significantly influenced by whether a conscious experience occurred (Figs 2A, 2B, and S4 and S2 Table; for the main effect of experience also see S4 Table). Specifically, the positive association between *delta* power and subjective sleep depth was stronger in NCE compared to CE and CEWR reports, most prominently in a right-frontal electrode cluster (*interaction β* = [−0.336, −0.302]). Conversely, the negative associations of *gamma* power (*interaction β* = [0.380, 0.467]) and the *power* ratio (*interaction β* = [0.232, 0.333]) with perceived sleep depth were attenuated when participants reported a conscious experience. In particular, significant interaction effects were localized to occipital electrodes for *gamma* power and to occipital and bilateral centro-frontal clusters for the *gamma/delta* ratio. Importantly, these interaction effects were specific to subjective sleep depth and did not extend to subjective sleepiness (Fig 2C; S3 Table; also see S5 Table), suggesting a distinct modulation of perceived sleep depth by the presence of ongoing mental activity.

Taken together, these findings suggest that increased cortical activation during N2 sleep is linked to a shallower subjective experience of sleep and reduced sleepiness upon awakening. Moreover, they show that the presence and quality of conscious experience during sleep modulate the mapping between brain activity and perceived sleep depth.

## The relationship between conscious experience and subjective sleep depth

We next examined how the presence and nature of conscious experience during sleep were related to subjective sleep depth and sleepiness upon awakening. Perceived sleep depth was significantly higher following reports of conscious experiences (CE+CEWR) compared to NCE ($p = 0.01307$; $β = −0.182$, CI = [−0.326, −0.038]), whereas no significant differences emerged for subjective sleepiness ($p = 0.25967$; S5 Fig; S4 and S5 Tables).

These results challenge the prevailing notion that dreamless sleep corresponds to the deepest sleep. One possible explanation is that not all NCE reports reflect true unconsciousness, but rather the absence of reportable content. Indeed, prior work has proposed the existence of minimal forms of consciousness during sleep, such as a mere *sense of presence* or the passage of time, that may elude common post-awakening questioning approaches [13]. To test this, we asked participants who reported NCEs to indicate whether they believed they had emerged from a minimal conscious state with a *sense of presence* (CESP) or from a state of complete *unconsciousness* (UNC). CESP was defined as a condition in which participants reported a mere awareness of being 'present' or of time passing, despite the absence of any specific dream content. Strikingly, only half of NCE reports were classified as UNC (50.15% ± 37.40%), while the remaining half reflected a contentless *sense of presence* (CESP; sign rank test, $p = 0.330$).

To further examine how the nature of conscious experience influenced subjective sleep quality, we repeated the previous analyses using a more fine-grained, four-level categorization: CE, CEWR, CESP, and UNC. This extended classification revealed systematic differences in perceived sleep depth across report types (Fig 3B; S4 Table). The lowest sleep depth ratings followed minimal conscious experiences (CESP), while both dream experiences (CE: $|β| = 0.552$, CI = [0.334, 0.770]; CEWR: $|β| = 0.553$, CI = [0.365, 0.742]) and complete unconsciousness (UNC: $|β| = 0.593$, CI = [0.322, 0.865]) were associated with significantly higher ratings (all $q < 0.001$, FDR correction). In contrast, subjective sleepiness showed a distinct pattern (Fig 3C; S5 Table): UNC reports were associated with the highest levels of sleepiness upon awakening, whereas all three categories involving some form of conscious mentation were linked to lower sleepiness (CE: $q = 0.0152$, $|β| = 0.289$, CI = [0.105, 0.472]; CEWR: $q = 0.0030$, $|β| = 0.311$, CI = [0.147, 0.474]; CESP: $q = 0.0366$, $|β| = 0.330$, CI = [0.089, 0.571]).

These results highlight that perceived sleep depth and sleepiness follow distinct trajectories across different states of consciousness. While both contentful dreaming and unconscious sleep are perceived as deeper than minimal conscious

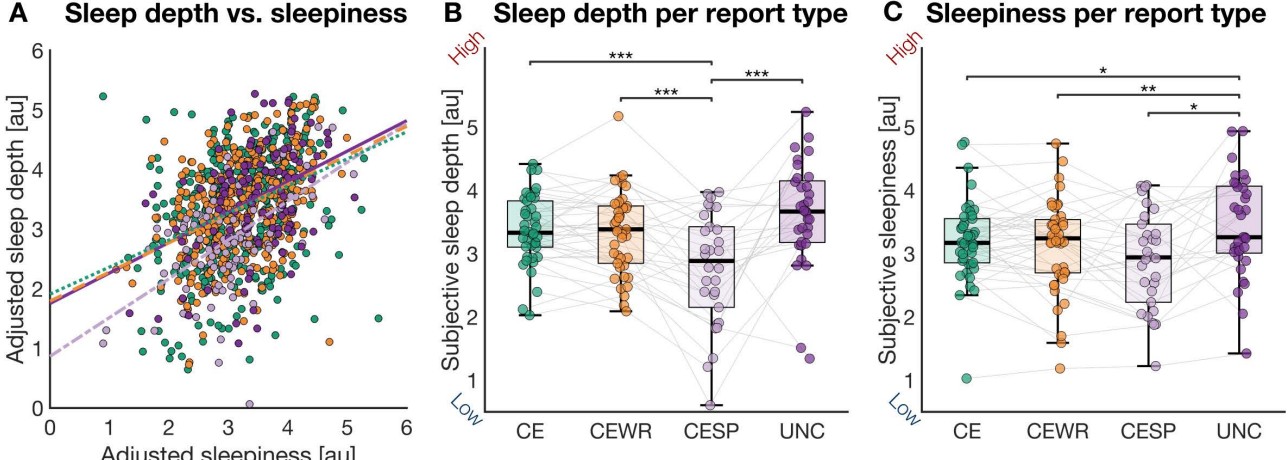

**Fig 3. Relationship between conscious experience, subjective sleep depth, and sleepiness. (A)** Correlation between subjective sleep depth and sleepiness, shown separately for the report types CE (Spearman $\rho = 0.38$), CEWR ($\rho = 0.40$), CESP ($\rho = 0.51$), and UNC ($\rho = 0.36$). Displayed values are adjusted for experiment, night, time of the night, and participant. **(B, C)** Mean subjective sleep depth (B) and sleepiness (C), each rated on a 5-point Likert scale, as a function of report type. Each dot represents the average (adjusted) score for an individual participant. Statistical significance was assessed via GLME models with FDR correction across multiple comparisons: *$q < 0.05$, **$q < 0.01$, ***$q < 0.001$. Displayed values are adjusted for experiment, night, and time of the night. In box plots, the box spans the interquartile range (IQR), the horizontal line indicates the median, and whiskers extend to the most extreme values within 1.5×IQR. Gray lines link data from the same participant across conditions. Abbreviations: CE, conscious experience (green), CEWR, conscious experience without recall of content (orange), CESP, conscious experience with sense of presence (light purple), and UNC, unconsciousness (dark purple). The data and code used to generate this figure are openly available (https://doi.org/10.5281/zenodo.15372992).

states limited to a sense of presence, any form of conscious experience, whether contentful or lacking any reportable content, was associated with reduced sleepiness upon awakening as compared to full unconsciousness.

## The link between experience phenomenology and subjective sleep depth

Next, we investigated whether specific phenomenological features of conscious experiences were associated with variations in subjective sleep depth and sleepiness. To this end, we applied principal component analysis (PCA) to six self-rated experiential dimensions: subjective duration (expressed in seconds and log-transformed), perceptual versus thought-like content, vividness, bizarreness, emotional intensity, and awareness of the dreaming state (Likert scales from 1 to 5). Among the resulting components, we retained the first two PCs based on explained variance (Fig 4A). The first component (PC1; 31.5% of total variance) showed high loadings on features typical of rich, immersive REM sleep dreams, including duration, vividness, perceptual richness, bizarreness, and emotional intensity. This pattern suggested a dimension reflecting the sensory richness and immersive quality of dream experiences, which we termed '*perceptual immersion*'. The second component (PC2; 20.0% of total variance) was characterized by opposite loadings on vividness and on thought-like qualities and meta-awareness of dreaming, indicating a tendency toward more reflective, less perceptual experiences. We referred to this dimension as '*reflective thought*'.

Statistical analyses showed that *perceptual immersion* was positively associated with subjective sleep depth ($\beta = 0.277$, CI = [0.208, 0.346], $p < 0.00001$), whereas *reflective thought* was negatively associated with subjective sleep depth ($\beta = -0.195$, CI = [−0.287, −0.102], $p = 0.00004$). In other words, richer, more immersive dream experiences were linked to a stronger perception of deep sleep (S6 Table). Notably, this association held even when participants could not recall the specific content of their dreams (Fig 4D; S4 Table). Indeed, subjective sleep depth was significantly higher following conscious experiences without recall of content (CEWR) in which participants reported the impression of having

**Fig 4. Relationship between phenomenological experience and subjective sleep depth and sleepiness. (A)** A principal component analysis (PCA) was performed on subjective ratings of dream duration, vividness, perceptual (vs. thought-like) content, bizarreness, emotional intensity, and awareness of the dreaming state, to reduce dimensionality. The first two components, PC1 ("perceptual immersion") and PC2 ("reflective thought"), were selected for further analysis. **(B)** Associations between PC1 (left) and PC2 (right) and subjective sleep depth. **(C)** Associations between PC1 (left) and PC2 (right) and subjective sleepiness. **(D)** Differences in subjective sleep depth (left) and sleepiness (right) for white dreams associated with the impression of a rich, immersive experience (rCEWR) vs. those without such an impression (sCEWR). In panels B–D, values are adjusted for experiment, night, and time of the night. Each dot represents a participant's mean rating. All GLME models included participant identity as a random effect and experiment, night, and time of night as fixed effects. In box plots, the box spans the interquartile range (IQR), the horizontal line indicates the median, and whiskers extend to the most extreme values within 1.5 × IQR. The data and code used to generate this figure are openly available (https://doi.org/10.5281/zenodo.15372992).

forgotten a rich, complex dream (*rich* CEWR, rCEWR) compared to those without such an impression (*simple* CEWR, sCEWR; β = −0.259, CI = [−0.459, −0.058], p = 0.01164). Interestingly, sCEWR made up the majority of CEWR reports (70.39 ± 30.73%; sign rank test, p < 0.0001), consistent with the hypothesis that CEWR may often (though, not always) reflect a failure to encode low-salience experiences in memory [14].

PLOS Biology

Analyses of the relationships between subjective sleepiness and features of conscious experiences revealed only a modest positive association with *perceptual immersion* ($\beta$ = 0.079, CI = [0.018, 0.140], $p$ = 0.01129), no effect of *reflective thought* ($\beta$ = −0.032, CI = [−0.109, 0.044], $p$ = 0.40777; Fig 4C), and no significant differences between rCEWR and sCEWR ($\beta$ = −0.096, CI = [−0.265, 0.074], $p$ = 0.26889; S5 and S6 Tables).

Taken together, these results indicate that the immersive quality of dream experiences is closely associated with perceived sleep depth, independent of whether specific content is recalled upon awakening. In contrast, reflective or abstract mental content correlated with shallower perceived depth, whereas dream phenomenology showed only a limited relationship with subjective sleepiness.

## Temporal dynamics of subjective sleep depth across the night

Finally, we examined how subjective sleep depth and sleepiness evolved over the course of the night by modeling their relationship with absolute clock time relative to midnight (Fig 5A). To account for shared variance between the two subjective measures, each model included the other measure as a covariate (i.e., sleepiness was included as a fixed-effect in the model predicting sleep depth, and vice versa; S7 Table). This analysis revealed a significant increase in perceived sleep depth as the night progressed ($\beta$ = 0.127, CI = [0.103, 0.151], $p$ < 0.00001). In contrast, subjective sleepiness did not follow a linear trajectory ($\beta$ = −0.012, CI = [−0.032, 0.008], $p$ = 0.23975), but instead displayed a nonlinear pattern: it increased during the first part of the night, peaked between 3 and 4 AM, and declined thereafter. This pattern is consistent with the expected circadian modulation of sleep propensity, which typically peaks in the early morning hours before decreasing (e.g., [15]). Notably, this divergence in time course revealed a dissociation between subjective sleep depth and sleepiness, particularly in the second half of the night. To test this statistically, we fitted a GLME model to post-4 AM data and found a significant interaction between time and measure (sleep depth versus sleepiness), indicating that sleepiness increased less steeply than perceived sleep depth ($\beta$ = −0.166, CI = [−0.277, −0.056], $p$ = 0.00325).

We next examined time-dependent changes in EEG activity that may explain the observed dissociation between subjective sleep depth and sleepiness (Fig 5C; S7 Table). *Gamma* power (25–50 Hz) exhibited a temporal pattern mirroring that of subjective sleepiness: it decreased early in the night, reached a nadir around 3–4 AM, and subsequently increased ($\beta$ = −0.007, CI = [−0.018, 0.004], $p$ = 0.20634). In contrast, *delta* power (0.5–4 Hz), a classical objective marker of homeostatic sleep pressure, showed a gradual linear decline consistent with known reductions in sleep pressure over the course of the night ($\beta$ = −0.072, CI = [−0.090, −0.054], $p$ < 0.0001). The *ratio* of *gamma* to *delta* power increased significantly with time ($\beta$ = 0.065, CI = [0.044, 0.086], $p$ < 0.0001), suggesting a shift in spectral balance toward higher-frequency, wake-like activity.

Overall, these neurophysiological dynamics are consistent with the well-established decline of both homeostatic and circadian sleep drives as one approaches the final morning awakening [16]. However, they do not explain the observed increase in subjective sleep depth. Indeed, subjective depth continued to rise during the second half of the night, precisely when both physiological sleep pressure and sleepiness diminished. Based on these changes, one would expect subjective sleep depth to decrease (as it happens for sleepiness) rather than increase. This apparent dissociation suggests the involvement of additional factors. We therefore tested whether changes in dream phenomenology could account for this discrepancy. Analyses revealed that *perceptual immersion* in dream experiences increased significantly with time ($\beta$ = 0.103, CI = [0.053, 0.154], $p$ < 0.0001; Fig 5B; S7 Table), indicating that dreams became progressively richer and more immersive as the night unfolded (also see S6 Fig). These findings support the view that immersive dreaming may help maintain the subjective sense of deep sleep, counteracting the decline in sleep propensity driven by homeostatic and circadian processes.

## Discussion

Contrary to the longstanding view that subjectively deep sleep is defined by unconsciousness and synchronized cortical activity dominated by high-amplitude slow waves, our findings demonstrate that the feeling of deep sleep can also

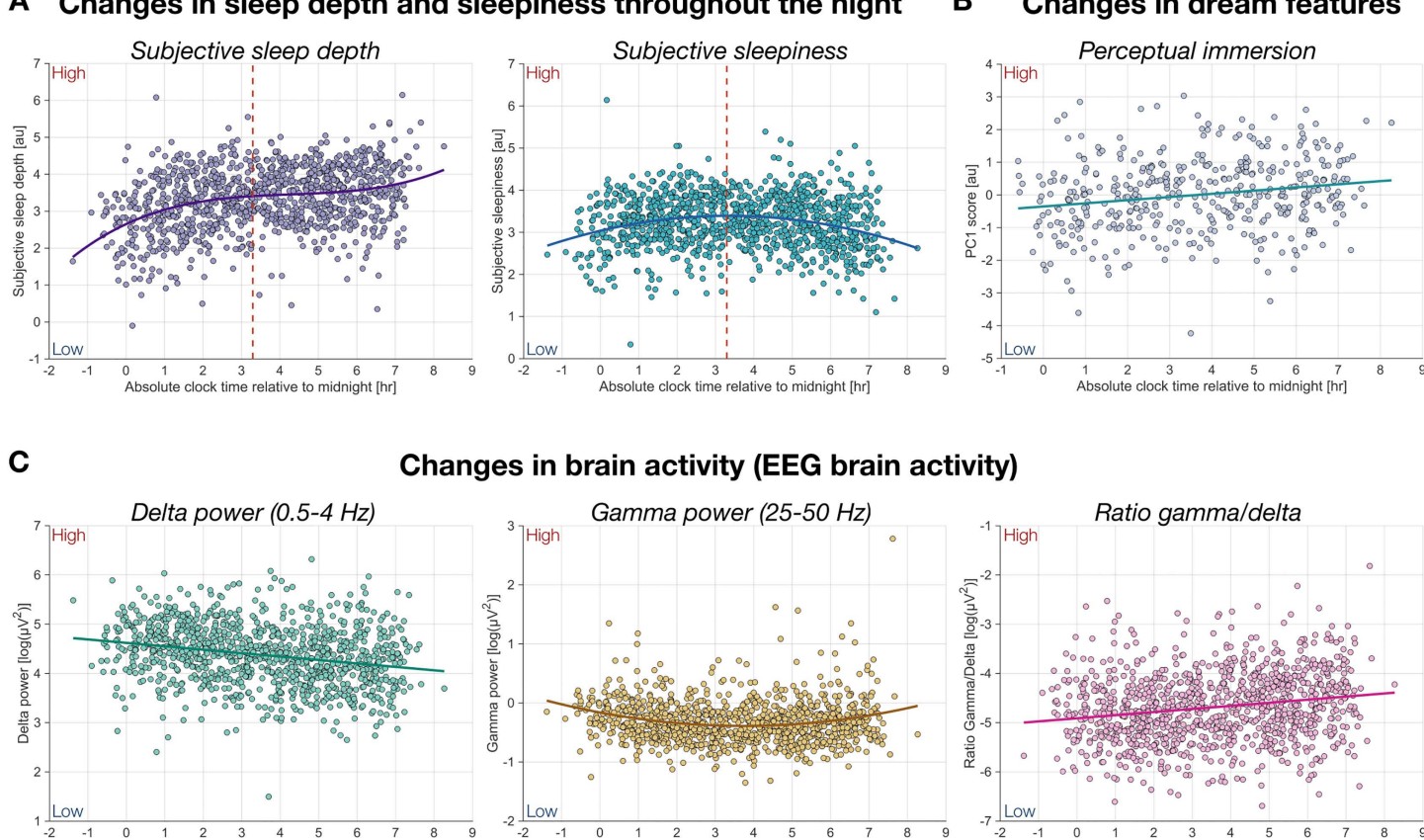

**Fig 5. Time-of-night effects on subjective experience and brain activity. (A)** Subjective sleep depth (left) and subjective sleepiness (right) as a function of clock time relative to midnight (0 hour). The red dashed line indicates the estimated peak in subjective sleepiness (based on model fit), occurring at 3:18 AM, after which sleepiness gradually decreased. **(B)** Dream immersiveness (PC1) plotted as a function of time of night. **(C)** EEG activity across the night: delta power (0.5–4 Hz; left), gamma power (25–50 Hz; middle), and the gamma/delta power ratio (right). All values are adjusted for experiment, night, and participant. In panel A, sleep depth is additionally adjusted for sleepiness, and vice versa, to account for shared variance. Each dot represents one observation ($N = 1,024$; $N = 427$ in panel B, where only CE reports with self-reported assessments were included). Curves represent the best-fitting polynomial model (linear, quadratic, or cubic) selected using the Bayesian Information Criterion (BIC). The data and code used to generate this figure are openly available (https://doi.org/10.5281/zenodo.15372992).

occur alongside increased cortical activation and vivid, immersive dreaming. Our study yielded three main insights. First, although wake-like high-frequency activity is typically associated with lighter subjective sleep, this relationship is modulated in the presence of conscious experience, so that wake-like cortical activation appears to be less disruptive to the feeling of deep sleep when it occurs alongside dreaming. Second, this shift is modulated by the immersive quality of dreams, as richer and more vivid experiences are more strongly associated with deeper perceived sleep. Third, such a modulation appears to sustain subjective sleep depth over the course of the night, contrasting the progressive physiological decline of sleep need and propensity. Together, these findings point to a previously unrecognized role of dreaming in shaping the perceived depth, and thus the subjective quality, of sleep.

Consistent with previous findings [8,17,18], we observed that increased high-frequency EEG power, as well as a higher high-to-low frequency power ratio, predicted lighter perceived sleep depth and lower subjective sleepiness upon awakening. In contrast, *delta* power, which reflects slow-wave activity [19], showed positive but nonsignificant associations with

sleep depth and sleepiness. Crucially, however, all associations between brain activity indices and subjective sleep depth (but not sleepiness) were significantly modulated by the presence of conscious experience. When participants reported having dreamt, the typical inverse relationship between high-frequency activity and perceived sleep depth was attenuated, and the positive association with *delta* power was similarly diminished. In other words, dream reports were linked to deeper subjective sleep even in the presence of relatively wake-like cortical activation patterns.

This decoupling of physiological and experiential markers of sleep depth suggests that dreaming may alter how brain activity translates into the perceived depth of sleep. These findings align with prior studies showing that dreaming can be accompanied by both decreased slow-wave activity and elevated high-frequency power, particularly in posterior cortical regions [4,10,20]. Given that high-frequency activity is generally associated with shallower sleep perception, one might expect dreaming, marked by increased cortical activation, to undermine the experience of deep sleep. Yet this is not the case: dreaming is a ubiquitous feature of healthy sleep and is not typically perceived as disruptive [21,22]. On the contrary, REM sleep, characterized by intense dream activity and cortical dynamics similar to wakefulness [6], is often rated as especially deep [8]. A similar association characterizes anesthetics such as ketamine, which produce states of deep subjective detachment alongside partial cortical activation and vivid, dream-like hallucinations [23,24]. Our findings extend this (apparent) paradox to N2 sleep, showing that even during this stage, immersive conscious experiences can offset the influence of increased cortical activation on the subjective experience of sleep depth.

Our results further revealed that the relationship between conscious experience and perceived sleep depth depends critically on the immersive quality of the experience. Subjective sleep depth was highest when participants awoke either from vivid, immersive dreams or from states lacking any conscious experience. By contrast, minimal forms of mentation, such as a mere feeling of presence or an awareness of time passing [13], were associated with the shallowest perceived depth. Crucially, this modulation by conscious experience did not require recall of specific dream content: participants who had no explicit memory of the dream content but reported the impression of having had a rich or immersive experience still rated their sleep as deeper. These findings align well with previous reports that individuals with *sleep state misperception* [25,26], who often report feeling awake during objective sleep, tend to exhibit less perceptual and more thought-like dreaming compared to good sleepers [8,11]. Thought-like mentation, being less immersive, may fail to reinforce the sensation of deep sleep, contributing to altered subjective sleep quality in this population.

In contrast to perceived sleep depth, subjective sleepiness followed a different pattern: its ratings were highest after unconscious sleep and lowest following any form of conscious experience. The increased sleepiness observed after unconscious sleep may reflect stronger post-awakening sleep inertia, a phenomenon associated with reduced cortical activation and higher slow-wave activity [17]. By contrast, dreaming has been linked to more activated cortical states [4], which could facilitate a smoother transition to wakefulness. Such transient activation may attenuate sleep inertia, leading to lower perceived sleepiness despite the concurrent feeling of deep sleep. This dissociation indicates that sleep depth and sleepiness reflect partially distinct subjective dimensions, shaped by different neural and psychological processes. The distinction became particularly evident when examining how these measures and brain activity indices evolved across the night. As expected, *delta* power, a well-established marker of homeostatic sleep pressure, declined steadily, reflecting the gradual dissipation of sleep need [16,27]. Subjective sleepiness followed a partially different, nonlinear pattern: it increased during the early part of the night and then decreased toward morning. This pattern, which was mirrored by *gamma* power, appears consistent with circadian influences on arousal and alertness [15]. Crucially, despite this dual decline in both homeostatic (*delta* power) and subjective (sleepiness) indicators of sleep propensity, subjective sleep depth did not decrease. Instead, it kept increasing across the night.

The divergence between physiological sleep pressure and perceived sleep depth may be explained, at least in part, by concurrent changes in dreaming. Extending prior reports that N2 dreams may become more REM-like across the night [28,29], we found that immersive dream experiences became increasingly frequent and vivid as the night progressed and were strongly associated with greater perceived depth. These findings suggest that immersive dreaming may actively

sustain or enhance the subjective perception of deep sleep, even as physiological sleep need wanes and cortical activity becomes more wake-like. Rather than simply tracking homeostatic and circadian signals, perceived sleep depth may thus be critically shaped by the processes that generate and structure dream experiences.

Overall, our findings challenge current models linking EEG markers to sleep perception and highlight a possible role of dreaming in shaping the subjective depth and continuity of sleep. We propose that the dream state functions as a richly immersive simulation that sustains the sensation of deep sleep, even in the face of fluctuating cortical dynamics or declining physiological sleep pressure. From an evolutionary standpoint, this capacity of dreams to preserve the feeling of being asleep may represent a functional adaptation. Fluctuations in cortical activation have been linked to sleep-dependent processes such as memory consolidation and emotion regulation [30], and dreams have been suggested to reflect these processes [31]. Yet, they may be more than mere epiphenomena: immersive dreams could buffer the arousing effects of such neural events, enabling a stable sense of deep sleep despite underlying neurophysiological variability. On a similar line, dreams have long been thought to promote sensory disconnection and protect sleep against external disturbances [3,32]. Freud famously referred to dreams as the "*guardians of sleep*" [33], and more recent accounts suggest that immersive dream experiences may attenuate the arousing power of external stimuli by incorporating them into ongoing mentation [34]. Our findings extend this view: immersive dreams may not only mitigate the impact of sensory input but also amplify the perception of sleep depth, acting as an 'internal scaffold' that helps the sleeper remain asleep. Thus, by engaging cortical networks in structured, internally generated activity, dreaming could serve a dual role: facilitating neural processes that rely on transient reactivations, such as memory and emotion processing, while simultaneously reinforcing the subjective perception of deep, undisturbed sleep in the presence of both internal and external perturbations.

Several limitations of this work, as well as implications for future research, warrant discussion. First, the present study is correlational and therefore cannot definitely establish a causal link between immersive dreaming and perceived sleep depth. Nonetheless, this interpretation aligns with prior evidence linking dreaming to sensory disconnection during sleep [34], and with reports associating altered dream phenomenology with poorer subjective sleep quality (e.g., [8]). Future work should directly test potential causal relationships, for example, by manipulating dream occurrence or complexity through controlled sensory-stimulation paradigms. Second, the use of the serial awakening paradigm inevitably fragments sleep and may influence both dream content and perceived sleep depth across the night. This limitation is particularly evident in multi-stage awakening protocols, which typically require numerous awakenings per night and can substantially alter sleep architecture (e.g., [8]). Such alterations may directly influence the processes under investigation, including both dream phenomenology and perceived sleep depth. For example, repeated interruptions of N3 early in the night may interfere with the normal dissipation of homeostatic sleep pressure, potentially biasing subjective reports obtained in later sleep periods. Similarly, repeated awakenings from REM sleep may alter REM pressure or affect dream recall and content later in the night. To mitigate these effects, we deliberately restricted awakenings to N2 sleep, a stage that offers sufficient variability in both physiology and conscious experience, while minimizing disruption of sleep-dependent processes in N3 and REM. Although this approach precluded direct cross-stage comparisons, it enabled an in-depth characterization of neural and experiential fluctuations within a single, well-defined physiological state. Future studies should extend this approach to other sleep stages, particularly N3 and REM, to assess whether the relationships between dream phenomenology, cortical activation, and perceived sleep depth generalize across the sleep cycle. Third, a related concern is the potential for carry-over effects from preceding sleep stages. As in all serial awakening studies, residual physiological or memory traces from prior REM or N3 episodes cannot be completely excluded. To minimize this possibility, participants were explicitly instructed to describe only their final experience immediately before the alarm sound (following procedures from [4,9,10]). In addition, we conducted supplementary control analyses quantifying the distance between each awakening and the preceding REM or N3 episode. These analyses revealed no significant effects on perceived sleep depth or dream phenomenology (S8 Table; also see S9 Table), suggesting that our findings are unlikely to reflect cross-stage contamination but rather capture intrinsic N2 dynamics.

If experimentally confirmed, a causal link between dreaming and subjective sleep depth or quality would have substantial clinical relevance. Disruptions in dream immersion and content have been observed in parasomnias and different forms of insomnia, and may contribute to reduced sleep satisfaction and perceived sleep depth [35]. For example, less immersive, thought-like dreams have been associated with the perception of shallow or insufficient sleep [26], while overly intense or dysphoric dreams, as in nightmare disorder, can degrade sleep quality by inducing arousals and sleep fragmentation [36]. As with many aspects of sleep physiology, optimal dream expression may lie within a balanced range, with deviations in either direction contributing to poor sleep experiences. Future work should determine whether targeted modulation of dream phenomenology, through behavioral, cognitive, or pharmacological approaches, can enhance sleep quality and help restore the felt depth of sleep in vulnerable populations. Moreover, while the present study delineates distinct subjective states during sleep, future investigations will be needed to further refine their underlying neural characterization. In particular, it will be important to examine how slowly evolving changes in global sleep dynamics interact with faster fluctuations and discrete microevents, such as sleep spindles or K-complexes, to shape moment-to-moment conscious experience and perceived sleep depth.

## Materials and methods

### Participants

The present study draws on data from two independent experiments (Experiment #1 and Experiment #2; see below) that both employed overnight high-density EEG recordings in combination with a serial awakening paradigm targeting N2 sleep. Both experiments used identical inclusion and exclusion criteria.

Potential volunteers between 18 and 40 years of age were recruited through advertisements and by word of mouth. The following exclusion criteria were applied: irregular bed and rise times; less than 6 hours or more of 9 hours of sleep per night; low subjective sleep quality (*Pittsburgh Sleep Quality Index* [37], score > 8); extreme chronotype (*Morningness-eveningness questionnaire* [38], score > 70 or < 30); excessive daytime sleepiness (*Epworth Sleepiness Scale* [39], score > 14); excessive anxiety (*Beck's Anxiety Inventory* [40], score > 18) or depressive symptoms (*Beck's Depression Inventory* [41], score > 14); use of medications known to significantly affect sleep patterns, brain functioning, or behavior; current or past diagnosis of sleep, cognitive, neurological, psychiatric, or neurodegenerative disorders; history of alcohol or illicit drug use in the last 6 months; pregnancy or breastfeeding; the presence of contraindications to MRI examination (e.g., pacemaker, metallic foreign bodies, etc.). The study was conducted in accordance with the principles expressed in the Declaration of Helsinki and was approved by the local *Joint Ethical Committee for Research of the Scuola Superiore Sant'Anna and the Scuola Normale Superiore* (resolutions 21/2020 and 22/2020). Written informed consent was obtained from all participants. Volunteers received monetary compensation for their participation in the study.

A total of 29 volunteers were screened for participation in Experiment #1. One was excluded for not meeting the inclusion criteria, while two withdrew from the study. Additionally, two participants were excluded after one night due to an inability to fall asleep during the experimental sessions. For Experiment #2, 33 volunteers were screened. Four did not meet the inclusion criteria and were excluded. Of the remaining participants, three were excluded after two nights and one after a single night due to difficulties falling asleep during the experimental sessions.

### Experimental procedure

In both experiments, participants completed four nonconsecutive overnight experimental sessions. Experiment #1 involved the completion of a 2-hour task session before sleep. During this time, participants practiced with a visual, auditory, or tactile task or a set of standardized activities. Instead, in Experiment #2, single brief visual, auditory, or vibrotactile stimulations were administered during consolidated N2 sleep, with awakenings induced shortly after the occurrence of a *K*-complex, that is, a large-amplitude evoked slow-wave with peak frequency at ~1 Hz. Sham trials, in which awakenings

were not preceded by stimulation, were also performed. The analyses described in this work combined all awakenings performed in Experiment #1 and sham awakenings obtained from Experiment #2. Detailed procedures for both experiments, as well as methods not directly relevant to the present report, are provided in the S1 Text.

During an initial screening interview, all participants completed several questionnaires and cognitive tests in addition to those described in the 'Participants' section, including Edinburgh Handedness Inventory [42], Attitude Towards Dreaming scale [43], Vividness of Visual Imagery Questionnaire [44], State-Trait Anxiety Inventory [45], Babcock Story Recall test [46], and Rey–Osterrieth Complex Figure test [47]. Then, the eligible participants received a detailed description of the experiment and the dream questionnaire used to assess subjective experiences following each awakening (see below). At the end of the screening interview, volunteers received an actigraphy (MotionWatch 8, CamNtech) and were asked to wear it on their nondominant wrist to monitor rest-activity patterns until completion of the fourth and last experimental night. The actigraphic data were checked before each experimental night to ensure that participants had regular bed and rise patterns and adequate sleep in the days leading to the experiment.

Throughout each night, brain activity was recorded using high-density EEG (256 electrodes; eego mylab with waveguard original caps, AntNeuro) with an online sampling rate of 500 Hz. EEG signals were referenced online to the Z12 electrode, positioned roughly posterior to Cz in accordance with the 10–20 system topology. Electrode impedance was brought below 20 kΩ, and electrode position was digitized (xensor, AntNeuro) before the beginning of the experiment. The polysomnographic setup also included: vertical and horizontal electrooculography (EOG), chin electromyography (EMG), electrocardiography (ECG), and a respiratory belt.

During the experimental nights, sleep staging was performed online based on the standard criteria outlined in the American Academy of Sleep Medicine (AASM) Scoring Manual v. 2.6 [48]. All experimenters received training in AASM-based scoring and followed detailed procedural manuals to ensure full consistency across sessions. Awakenings were induced using a 1.5 s computerized alarm sound after at least 15 min from sleep onset and 5 min of stable N2 sleep. Awakenings in other sleep stages were avoided to minimize possible alterations to sleep structure and sleep-related processes. The awakening criteria and procedures were identical across Experiments #1 and #2. To minimize possible cross-stage contaminations and memory carry-overs, awakenings within 10 min from the end of a REM sleep period were also avoided. Experimenters were instructed to perform up to 4 awakenings per sleep cycle, with a target of 8–12 awakenings per night. The experimental sessions concluded around 7 AM. Only data from participants who completed all four experimental nights were analyzed.

## Dream questionnaire

Upon each awakening, participants were asked a series of prerecorded questions regarding their subjective experience. A detailed manual explaining all definitions and questions included in the questionnaire was read by the experimenters to all participants during the screening interview (an English translation of the manual is provided in S10 Table). The experimenters ensured that all questions were clearly understood. Moreover, volunteers were instructed to practice answering the questionnaire each morning at home for 1 week to familiarize themselves with the questions and scales. Before the start of each experiment, the experimenters reviewed all the questions and their meanings with the participants.

During the experiment, a custom-made MATLAB code written with PsychToolbox-3 [49–51] was used to play the questionnaire and record the participants' responses. First, participants were asked to describe what was going through their minds just before they woke up. According to their responses, reports were classified into one of the following three categories: 'conscious experience' (CE), if the report described some kind of contentful subjective experience including thoughts, images, sensations, or emotions; 'conscious experience without recall of content' (CEWR), if the report indicated the impression of having dreamt but no specific contents were recalled (so-called 'white dream'); 'no conscious experience' (NCE) if the report indicated no subjective content of experience. Depending on the type of report provided upon awakening, participants were asked a different follow-up question to further characterize their experience. For CE reports,

they were asked to estimate the length of the conscious experience in minutes. For CEWR reports, participants indicated whether they had the impression of having forgotten a *rich* and detailed experience (rCEWR) or had no such impression (*simple* CEWR, or sCEWR). Finally, for NCE reports, participants were asked whether they had experienced a sense of presence, defined as a feeling of being alive and/or perceiving the present moment or the passage of time prior to awakening [13]. Reports affirming this were labeled as *conscious experience with a sense of presence* (CESP), while those reporting no such impression were labeled as *unconscious* (UNC). Thus, CESP identifies cases of minimal, contentless awareness, akin to *mind blanking* during wakefulness [52]. Of note, all responses were reevaluated offline to confirm the first-level classification of the report. The second-level information was removed from re-classified reports, which included four CE (0.9%) and three CEWR (0.8%) reports. A Friedman test was used to compare the relative proportions of the different report types.

All participants were then asked to use 5-point Likert scales to indicate how deep was their sleep before the alarm sound and how tired and sleepy they felt upon waking. Moreover, they had to indicate whether they perceived any sensory stimuli just before the alarm sound. Participants who reported a conscious experience were asked to indicate the subjective duration of the experience (in seconds, minutes, or hours) and to use 5-point Likert scales (1–5) to characterize their experience according to the following dimensions: vividness (from low to high), percentual versus thought-like content (from exclusively thought-like to exclusively perceptual), bizarreness (from low to high), degree of relatedness to events encountered during wakefulness (from absent to strong), emotional valence (from strongly negative to strongly positive), emotional intensity (from low to high), awareness of being into a dream (from no to high), and voluntary control over the dream (from no to full). Moreover, they had to indicate the type of sensory contents that were present in the experience (yes/no: visual, auditory, tactile, olfactory, and gustatory), and their degree of confidence in the accuracy and reliability of their report (from low to high). In order to obtain a phenomenological characterization of conscious experiences, a PCA was applied to six self-reported features, including experience duration (log-transformed), bizarreness, vividness, emotional intensity, degree of perceptual rather than thought-like content, and degree of awareness of the dreaming state. Each feature was normalized (*z*-scored) prior to PCA.

### Whole-night sleep scoring

Whole-night recordings were offline sleep scored using the *U-Sleep v2.0* automatic algorithm to obtain general descriptive statistics regarding sleep structure in each night, participant, and experiment (model version 2.0, 30-s epochs) [53]. The following channels were used: L3L (roughly corresponding to F3), L3A (C3), L6A (P3), L16Z (O1), R3R (F4), R3A (C4), R6A (P4), R16Z (O2), and bipolar vertical (VEOGR, R1Z) and horizontal (L1G, R1G) EOG. The data were filtered using a low-pass filter at 0.3 Hz and a high-pass filter at 35 Hz, downsampled to 128 Hz, and re-referenced to the contralateral mastoids (L4H, R4H). One experimenter (A.M.) manually rescored a subset of seven randomly selected nights to verify the algorithm's accuracy, confirming a high concordance between manual and automated scoring (average F1 score: 0.74 ± 0.07). Given this strong agreement and the prior validation of *U-Sleep*, no further large-scale manual validation was deemed necessary.

### EEG data preprocessing

Recordings were segmented into 362-s-long epochs ending 2 s after the onset of the alarm sound. All segments were assessed offline through visual inspection by one trained experimenter (A.M.) to verify whether the awakening occurred from N2 sleep (as defined according to AASM *2.6* scoring rules). For this evaluation, we selected derivations corresponding to F3, C3, P3, and O1, all referenced to the right mastoid channel. Bipolar EOG and EMG channels were also assessed. The EEG signals were high-pass filtered at 0.3 Hz and low-pass filtered at 35 Hz, while the EMG signal was high-pass filtered at 10 Hz and low-pass filtered at 100 Hz. Only awakenings confirmed to have occurred from NREM sleep entered the preprocessing procedure.

Retained data segments were preprocessed using a custom, semi-automated pipeline developed in MATLAB (The Matworks) and based on the EEGLAB toolbox (version 2021.0). Line noise was removed using the *ZapLine plus* toolbox [54,55], while noisy channels were identified and excluded using the PREP pipeline's *findNoisyChannels* wrapper function [56]. Artifacts were reduced via independent component analysis (ICA; *runica* function, [57]). For this, we first created a copy of the raw, unfiltered data [58]. All data epochs recorded during the same night were low-pass filtered at 45 Hz (*pop_eegfiltnew* function), high-pass filtered at 0.5 Hz, and then concatenated. Channels previously identified as bad were excluded from the ICA procedure. Moreover, a PCA was applied to reduce the number of extracted components to 150. The *ICLabel* plugin was applied to the extracted components [59]. Those components classified as 'brain' with a probability above 25%, and those classified as noise for which brain was the second most probable label and with a brain-to-noise ratio above 0.80 were retained. Selected components were confirmed or modified after visual inspection by expert scorers. Finally, ICA weights and transformation matrices were applied to the original, unfiltered recordings, and artifactual components were removed. Bad channels were interpolated using spherical splines (*pop_interp* function).

## EEG data analysis

EEG signal power was quantified within frequency bands previously shown to be associated with consciousness and the perception of sleep depth [4,8]: *delta* (slow-wave activity, 0.5–4.0 Hz), and *gamma* (high-frequency activity, 25–50 Hz). In addition, we computed the *gamma*/*delta* power *ratio* as an index of the balance between high- and low-frequency activity. To prepare the data, EEG recordings were first detrended using the *robust detrend* function [60] with a second-order polynomial to remove slow DC drifts. Data were then segmented into 120-s epochs immediately preceding the alarm onset. For each epoch, power spectral density (PSD) was estimated using Welch's method (MATLAB *pwelch* function) with 4-s Hamming windows and 90% overlap. Power within each frequency band was calculated by integrating the PSD over the respective range using the trapezoidal method.

## Statistical analyses

Across both experiments, the combined dataset included 1,137 awakenings, of which 1,024 were retrospectively confirmed to have occurred from N2 sleep. GLME models were used to examine the relationships linking neural, physiological, and phenomenological variables to subjective sleep depth and subjective sleepiness (see Table 1). All models included *experiment,* and *night* as categorical fixed effects, *time* of the night (relative to midnight) as a continuous fixed factor, and *participant* as a random effect. Of note, we also tested models including *time since lights-off* instead of absolute clock time, but results were highly consistent with those using *absolute time*, so only the latter are reported here. The relative influence of the different experimental protocols on the main outcome variables is summarized in S11 Table.

To identify the EEG correlates of subjective sleep depth (Fig 1A), and subjective sleepiness (Fig 2), we applied channel-wise GLME models. Separate models were run for each brain activity index *j* (*delta* power, *gamma* power, and gamma/delta ratio), all of which were log-transformed to improve distribution normality (models #1, #3). Moreover, to determine whether the predictive role of brain activity on subjective sleep depth or sleepiness was modulated by the presence or absence of conscious experience (*exp*), additional models were applied to explore the potential interaction between these predictors (models #2, #4). Wald statistics (i.e., squared fixed effect estimate divided by its standard error) were computed for each model. To correct for multiple comparisons across electrodes, a cluster-mass permutation approach was used [20,61]. Specifically, the predicted variable was shuffled 5,000 times within each participant and experiment. For each permutation, the same model was applied, and spatially contiguous electrodes with $p < 0.005$ (cluster-forming threshold) and consistent effect direction were grouped into clusters. To minimize the risk of false positives, a minimum cluster size of three spatially contiguous electrodes was applied to the topographic maps. The cluster-mass statistic was defined as the sum of Wald statistics within each significant cluster. For each permutation, the

**Table 1. Summary of generalized linear mixed-effects (GLME) models used in the study.** Each row describes a GLME model implemented to test specific hypotheses, as outlined in the main text. The "Formula" column specifies the model structure, including fixed effects and a random intercept for participant identity. The "Predictor of interest" column highlights the primary variable examined in each model. The "Fig." column indicates the figure panel(s) in which the corresponding results are shown. In the formulas: experiment denotes the experiment from which each data point originates (categorical; two levels); night refers to the experimental night (categorical; four levels); time refers to the time of the night relative to midnight (continuous); brain_activity represents the EEG measure of interest (delta power, gamma power, or the gamma/delta ratio); exp. refers to the presence or absence of conscious experience (CE+CEWR vs. NCE); report_type refers to categorical experience types (e.g., CE, CEWR, CESP, and UNC); PC refers to the two principal components derived from phenomenological ratings. All models included experiment, night, and time as fixed effects and participant as a random effect.

| N | Formula | Predictor of interest | Fig. |
|---|---|---|---|
| #1 | $sleep\_depth \sim 1 + brain\_activity_j + experiment + night + time + (1|participant)$ | Brain activity | 2A, 2B |
| #2 | $sleep\_depth \sim 1 + exp.*brain\_activity_j + experiment + night + time + (1|participant)$ | Interaction term | 2A |
| #3 | $sleepiness \sim 1 + brain\_activity_j + experiment + night + time + (1|participant)$ | Brain activity | 2C |
| #4 | $sleepiness \sim 1 + exp.*brain\_activity_j + experiment + night + time + (1|participant)$ | Interaction term | 2C |
| #5 | $sleep\_depth \sim 1 + report\_type + experiment + night + time + (1|participant)$ | Report type | 3B |
| #6 | $sleepiness \sim 1 + report\_type + experiment + night + time + (1|participant)$ | Report type | 3C |
| #7 | $sleep\_depth \sim 1 + pc_j + experiment + night + time + (1|participant)$ | Principal component | 4B |
| #8 | $sleepiness \sim 1 + pc_j + experiment + night + time + (1|participant)$ | Principal component | 4C |
| #9 | $sleep\_depth \sim 1 + sleepiness + experiment + night + time + (1|participant)$ | Time | 5A |
| #10 | $sleepiness \sim 1 + depth + experiment + night + time + (1|participant)$ | Time | 5A |
| #11 | $value \sim 1 + experiment + night + measure*time + (1|participant)$ | Interaction term | 5A |
| #12 | $brain\_activity_j \sim 1 + experiment + night + time + (1|participant)$ | Time | 5C |
| #13 | $pc_j \sim 1 + experiment + night + time + (1|participant)$ | Time | 5B |

maximum absolute cluster-mass was retained to build a null distribution. Observed clusters were considered significant if their cluster-mass exceeded the 95th percentile of the null distribution (corrected $p < 0.05$).

To investigate the association between subjective sleep depth or sleepiness and report type, separate models were fitted for each pairwise comparison of report categories (e.g., CE versus NCE; models #5, #6). This approach was chosen to avoid imposing assumptions about an ordinal or continuous relationship among categories, as current evidence does not support a unidimensional or equidistant hierarchy of consciousness levels during sleep. For comparisons across four report types (CE, CEWR, CESP, and UNC), false discovery rate (FDR) correction was applied to account for multiple comparisons across models and control the risk of false positive results [62].

To explore the association between dream phenomenology and subjective sleep depth or sleepiness (models #7, #8), separate models were applied for PC1 (*perceptual immersion*) and PC2 (*reflective thought*).

Finally, separate models were applied to investigate time-dependent changes in perceived sleep depth (model #9), sleepiness (model #10), dream immersiveness (model #13), and brain activity indices (model #12). Given the shared variance between sleep depth and sleepiness, each of the time-dependent models for these measures included the other variable as a fixed-effect to control for their mutual influence. In addition, to directly test whether subjective sleep depth and sleepiness exhibited significantly different trajectories across the night, we fit an additional model including both measures (*measure*: sleep depth, sleepiness) and their interaction with time in predicting reported values (*value*). This model was restricted to data collected after 4:00 AM, a time point identified as the peak of subjective sleepiness based on a second-order polynomial fit.

## Supporting information

**S1 Fig. Representative hypnogram from a single experimental night.** The trace illustrates the progression of sleep stages across the night of a participant in Experiment 1, with the x-axis indicating clock time (00 = midnight) and the y-axis denoting sleep stage (Wake, N1, N2, N3, and REM). Sleep scoring was performed using U-Sleep. Vertical dashed lines

mark experimentally induced awakenings delivered during N2 sleep. By restricting experimental interventions to N2, we maximized data collection within a single stage while minimizing potential alterations to the overall sleep architecture and reducing the risk of cross-stage carry-over or interaction effects.
(TIF)

**S2 Fig. Distribution of report types across participants and experiments.** Boxplots show the percentage of each report type for individual participants in Experiment 1 (single oblique line) and Experiment 2 (crossed lines). For each report type, rank-sum tests were used to assess differences between experiments; FDR-corrected $q$-values are reported above each pair of boxplots. No significant differences were observed across experiments. In box plots, the box spans the interquartile range (IQR), the horizontal line indicates the median, and whiskers extend to the most extreme values within $1.5 \times$ IQR. Report type abbreviations: CE, conscious experience; rCEWR, rich conscious experience without recall of content; sCEWR, simple conscious experience without recall of content; CESP, minimal conscious experience with a sense of presence; UNC, unconsciousness.
(TIF)

**S3 Fig. Topographic distribution of brain activity indices.** Topographic maps display the average spatial distribution of EEG indices in the 120 s preceding each awakening, separately for all reports, CE reports, CEWR reports, and NCE reports. Values were first averaged across awakenings within participants and then across participants, irrespective of experimental condition.
(TIF)

**S4 Fig. Neural correlates of subjective sleep depth as a function of conscious experience.** Topographic maps (top) show the interaction between brain activity and conscious experience (CE+CEWR versus NCE) for each brain activity index (the plots are identical to those shown in Fig 2). Wald statistics are projected onto the scalp, with white dots marking significant clusters after cluster-mass correction (corrected $p < 0.05$). The lower panel displays scatter plots illustrating the relationship between sleep depth and brain activity from a representative electrode (indicated on the topographic maps), shown separately for CE+CEWR (dark green) and NCE (purple). All GLME models included participant as a random intercept and experiment, night, and time of night (relative to midnight) as fixed effects; displayed values are adjusted accordingly.
(TIF)

**S5 Fig. Relationship between conscious experience and self-reported sleep depth and sleepiness.** Participants rated sleep depth (A) and sleepiness (B) on a 5-point Likert scale. Each dot represents the average score for an individual participant. Displayed values are adjusted for experiment, night, and time of night. Asterisks indicate statistical significance based on GLME results: *$p < 0.05$, **$p < 0.01$, ***$p < 0.001$. In box plots, the box spans the interquartile range (IQR), the horizontal line indicates the median, and whiskers extend to the most extreme values within $1.5 \times$ IQR. Gray lines link data from the same participant across conditions. Color coding: CE+CEWR (reports of conscious experience) = dark green; NCE (no conscious experience) = purple.
(TIF)

**S6 Fig. Time-of-night effects on subjective sleep depth and dream features.** (A) Subjective sleep depth as a function of clock time relative to midnight (0 hour) plotted separately for CE reports (left) and NCE reports (right). (B) Dream perceptual immersion (PC1; left) and reflective thought (PC2; right) components plotted as a function of time of night. All values are adjusted for experiment, night, and participant. In panel A, sleep depth is additionally adjusted for sleepiness scores to account for their shared variance. The left plot in panel B is the same as shown in Fig 5 of the main text, presented here for comparison with the other plots. Each dot represents one observation ($N = 432$ in panel A, left; $N = 228$ in panel A, right; $N = 427$ in panel B). Curves represent the best-fitting polynomial model (linear, quadratic, or cubic) selected using the Bayesian Information Criterion (BIC).
(TIF)

**S1 Table. Characteristics of the samples in the two experiments.** The table includes summary statistics regarding the participants' sleep structure (total sleep time and percentages of the different sleep stages) and the performed awakenings (absolute number and percentages of the different report types with respect to the total number of retained awakenings). For the age and the sleep structure indices, values are reported as mean ± standard deviation.
(PDF)

**S2 Table. Results of the GLME analyses examining the relationship between brain activity and subjective sleep depth at the electrode level.** The first two sets of models tested: (1) the main effects of neural predictors (delta power, gamma power, gamma/delta ratio), and (2) interactions between each neural predictor and report type (CE+CEWR versus NCE). The third and fourth sets of models examined the main effects of neural predictors separately for CE+CEWR (3) and NCE (4) reports. For each analysis yielding significant effects, the table presents results from the electrodes with the smallest and largest absolute β coefficients among those that reach significance. All models included experiment, night, and time of night as fixed effects, and participant as a random effect. Reported metrics include the number of observations (N Obs.), adjusted model $R^2$ ($R^2$ Adj.), likelihood-ratio test $p$-values (LRT p) comparing full and reduced models excluding the predictor of interest, differences in AIC and BIC (ΔAIC, ΔBIC), estimated regression coefficients (β) with 95% confidence intervals (CI low–high), and corresponding p-values. Positive ΔAIC or ΔBIC values indicate lower AIC/BIC for the full model.
(PDF)

**S3 Table. Results of the GLME analyses examining the relationship between brain activity and subjective sleepiness at the electrode level.** The first two sets of models tested: (1) the main effects of neural predictors (delta power, gamma power, and gamma/delta ratio), and (2) interactions between each neural predictor and report type (CE+CEWR versus NCE). Since no significant interactions were found, we did not examine the main effects of neural predictors separately for CE+CEWR and NCE reports. For each analysis yielding significant effects, the table presents results from the electrodes with the smallest and largest absolute β coefficients among those that reach significance. All models included experiment, night, and time of night as fixed effects, and participant as a random effect. Reported metrics include the number of observations (N Obs.), adjusted model $R^2$ ($R^2$ Adj.), likelihood-ratio test $p$-values (LRT p) comparing full and reduced models excluding the predictor of interest, differences in AIC and BIC (ΔAIC, ΔBIC), estimated regression coefficients (β) with 95% confidence intervals (CI low–high), and corresponding $p$-values. Positive ΔAIC or ΔBIC values indicate lower AIC/BIC for the full model.
(PDF)

**S4 Table. Results of GLME analyses testing whether subjective sleep depth differed across conscious states during sleep.** Each model compares a specific pair of report types (see 'Contrast' column) and includes experiment, night, and time of night as fixed effects, and participant as a random effect. Reported metrics include the number of observations (N Obs.), adjusted model $R^2$ ($R^2$ Adj.), likelihood-ratio test $p$-values (LRT p) comparing full and reduced models excluding the predictor of interest, differences in AIC and BIC (ΔAIC, ΔBIC), estimated regression coefficients (β) with 95% confidence intervals (CI low–high), and corresponding $p$-values. Positive ΔAIC or ΔBIC values indicate lower AIC/BIC for the full model. For comparisons involving the four-level classification (CE, CEWR, CESP, and UNC; Fig 2), false discovery rate (FDR) correction was applied, and the resulting q-value is shown in the final column. Statistically significant effects ($q < 0.05$) are indicated in bold.
(PDF)

**S5 Table. Results of GLME analyses testing whether subjective sleepiness differed across conscious states during sleep.** Each model compares a specific pair of report types (see 'Contrast' column) and includes experiment, night, and time of night as fixed effects, and participant as a random effect. Reported metrics include the number of

observations (N Obs.), adjusted model R² (R² Adj.), likelihood-ratio test *p*-values (LRT p) comparing full and reduced models excluding the predictor of interest, differences in AIC and BIC (ΔAIC, ΔBIC), estimated regression coefficients (β) with 95% confidence intervals (CI low–high), and corresponding *p*-values. Positive ΔAIC or ΔBIC values indicate lower AIC/ BIC for the full model. For comparisons involving the four-level classification (CE, CEWR, CESP, and UNC; Fig 2), false discovery rate (FDR) correction was applied, and the resulting q-value is shown in the final column. Statistically significant effects (*q* < 0.05) are indicated in bold.
(PDF)

**S6 Table. Results of the GLME analyses examining the relationship between the phenomenological features of conscious experience and subjective sleep depth or sleepiness.** All models included experiment, night, and time of night as fixed effects, and participant as a random effect. PC1: perceptual immersion; PC2: reflective thought. Reported metrics include the number of observations (N Obs.), adjusted model R² (R² Adj.), likelihood-ratio test *p*-values (LRT p) comparing full and reduced models excluding the predictor of interest, differences in AIC and BIC (ΔAIC, ΔBIC), estimated regression coefficients (β) with 95% confidence intervals (CI low–high), and corresponding *p*-values. Positive ΔAIC or ΔBIC values indicate lower AIC/BIC for the full model. Statistically significant effects (*p* < 0.05) are indicated in bold.
(PDF)

**S7 Table. Results of GLME analyses investigating the effect of time of the night on sleep depth, sleepiness, brain activity, and dream features.** All models include experiment, night, and time of night as fixed effects, and participant as a random effect. To account for shared variance between subjective sleep depth and sleepiness, each of the models exploring these variables included the other measure as an additional fixed effect. Reported metrics include the number of observations (N Obs.), adjusted model R² (R² Adj.), likelihood-ratio test *p*-values (LRT p) comparing full and reduced models excluding the predictor of interest, differences in AIC and BIC (ΔAIC, ΔBIC), estimated regression coefficients (β) with 95% confidence intervals (CI low–high), and corresponding *p*-values. Positive ΔAIC or ΔBIC values indicate lower AIC/BIC for the full model. Statistically significant effects (*p* < 0.05) are indicated in bold.
(PDF)

**S8 Table. Results of the GLME analyses testing the relationship between REM or N3 offset distance and subjective sleep depth or dream phenomenology.** Each model included experiment, night, and time of night as fixed effects and participant as a random intercept. PC1: perceptual immersion. Using the U-Sleep automatic staging, we computed the number of minutes separating each N2 awakening from the offset of the closest REM or N3 period within the same sleep bout. To minimize false detections, we excluded REM or N3 fragments containing fewer than three contiguous epochs and tied-rank transformed the distance values to reduce the influence of outliers. Sleep bouts containing no REM or N3 sleep were excluded from analysis. Reported metrics include the number of observations (N Obs.), adjusted model R² (R² Adj.), likelihood-ratio test *p*-values (LRT p) comparing full and reduced models, differences in AIC and BIC (ΔAIC, ΔBIC), estimated regression coefficients (β) with 95% confidence intervals (CI low–high), and corresponding *p*-values. Positive ΔAIC or ΔBIC values (i.e., lower AIC/BIC for the full model) indicate that including the stage distance factor improved model fit. No significant effects were identified.
(PDF)

**S9 Table. Results of the GLME analyses testing the relationship between cumulative REM sleep and subjective sleep depth or dream phenomenology.** Each model included experiment, night, and time of night as fixed effects and participant as a random intercept. PC1: perceptual immersion. Using the U-Sleep automatic staging, we computed for each N2 awakening the cumulative duration (in minutes) of preceding REM sleep. To minimize false detections, we excluded REM fragments containing fewer than three contiguous epochs, and tied-rank transformed the distance values to reduce the influence of outliers. Analyses were conducted both including all awakenings (top rows) and restricting to

awakenings with at least some preceding REM sleep (bottom rows). Reported metrics include the number of observations (N Obs.), adjusted model R² (R² Adj.), likelihood-ratio test $p$-values (LRT p) comparing full and reduced models, differences in AIC and BIC (ΔAIC, ΔBIC), estimated regression coefficients (β) with 95% confidence intervals (CI low–high), and corresponding $p$-values. Positive ΔAIC or ΔBIC values (i.e., lower AIC/BIC for the full model) indicate that including the REM amount factor improved model fit. Significant effects ($p < 0.05$) are shown in bold.
(PDF)

**S10 Table. Dream questionnaire.** Instructions for the volunteers, translated from Italian.
(PDF)

**S11 Table. Results of the GLME analyses examining the effect of the experimental protocol on subjective sleep depth, subjective sleepiness, and dream phenomenology.** Each model included experiment (Experiment #1, Experiment #2), night, and time as fixed effects and participant as a random intercept. PC1: perceptual immersion; PC2: reflective thought. Reported metrics include the number of observations (N Obs.), adjusted model R² (R² Adj.), the likelihood-ratio test $p$-value (LRT p) comparing the full model with a reduced model excluding experiment, the AIC and BIC differences (ΔAIC, ΔBIC), the estimated fixed effect coefficient (β; positive values indicate higher scores in Experiment #2 relative to Experiment #1), its 95% confidence interval (CI low–high), and the coefficient $p$-value. Positive ΔAIC or ΔBIC values (i.e., lower AIC/BIC for the full model) indicate that including the experiment factor improved model fit. Statistically significant effects ($p < 0.05$) are shown in bold.
(PDF)

**S1 Text. Additional experimental procedures.**
(PDF)

## Acknowledgments

The authors thank Mariachiara Amori, Giulia Avvenuti, Christina Bikeskou, Annkathrin Böke, Aikaterini Athina Bougoulia, Francesca Dalle Piagge, Isabella De Cuntis, Marie Degrave, Monica Di Giuliano, Tommaso Maccario, Lorenzo Meoli, Kim Mi Lande, Sıla Mutaf, Caterina Padreddii, Niccolò Pampaloni, Giorgia Procissi, and Leila Salvesen, for their help with data collection and preprocessing. The authors are also grateful to Guillaume Legendre for his comments and suggestions on an early version of the manuscript.

## Author contributions

**Conceptualization:** Giulio Bernardi.

**Data curation:** Adriana Michalak, Davide Marzoli, Francesco Pietrogiacomi, Valentina Elce, Giulio Bernardi.

**Formal analysis:** Giulio Bernardi.

**Funding acquisition:** Giulio Bernardi.

**Investigation:** Adriana Michalak, Davide Marzoli, Francesco Pietrogiacomi, Damiana Bergamo, Valentina Elce, Bianca Pedreschi, Giorgia Mosca.

**Methodology:** Alessandro Navari, Giacomo Handjaras, Giulio Bernardi.

**Project administration:** Giulio Bernardi.

**Software:** Giulio Bernardi.

**Supervision:** Michele Emdin, Emiliano Ricciardi, Giulio Bernardi.

**Visualization:** Valentina Elce, Giacomo Handjaras, Giulio Bernardi.

**Writing – original draft:** Adriana Michalak, Giacomo Handjaras, Giulio Bernardi.

**Writing – review & editing:** Adriana Michalak, Davide Marzoli, Francesco Pietrogiacomi, Damiana Bergamo, Valentina Elce, Bianca Pedreschi, Giorgia Mosca, Alessandro Navari, Michele Emdin, Emiliano Ricciardi, Giacomo Handjaras, Giulio Bernardi.

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
