## [Editor Report · Decision Letter 0]

11 Sep 2025

Dear Dr Bernardi,

Thank you for submitting your manuscript entitled "Immersive NREM dreaming preserves subjective sleep depth against declining sleep pressure" for consideration as a Research Article by PLOS Biology.

Your manuscript has now been evaluated by the PLOS Biology editorial staff as well as by an academic editor with relevant expertise and I am writing to let you know that we would like to send your submission out for external peer review. I should note that, while we are interested in your study, in principle, at this stage we are a bit on the fence about whether the insights provided offer the level of conceptual advance that we would need for publication. We will therefore be looking for enthusiasm and support from the reviewers to move forward with your study.

Before we can send your manuscript to reviewers, we need you to complete your submission by providing the metadata that is required for full assessment. To this end, please login to Editorial Manager where you will find the paper in the 'Submissions Needing Revisions' folder on your homepage. Please click 'Revise Submission' from the Action Links and complete all additional questions in the submission questionnaire.

Once your full submission is complete, your paper will undergo a series of checks in preparation for peer review. After your manuscript has passed the checks it will be sent out for review. To provide the metadata for your submission, please Login to Editorial Manager (https://www.editorialmanager.com/pbiology) within two working days, i.e. by Sep 16 2025 11:59PM.

Kind regards,

Luke

Lucas Smith, Ph.D.

Senior Editor

PLOS Biology

lsmith@plos.org

---

## [Decision Letter · Decision Letter 1]

3 Nov 2025

Dear Dr Bernardi,

Thank you for your patience while your manuscript "Immersive NREM dreaming preserves subjective sleep depth against declining sleep pressure" was peer-reviewed at PLOS Biology. It has now been evaluated by the PLOS Biology editors, an Academic Editor with relevant expertise, and we consulted three independent reviewers, although we have, to date, only received reports from two of them. We had been hoping to receive reviewer 3's comments last week, but have still not heard from them. However, we feel the two reviewers who have submitted comments already cover the relevant expertise, and after discussion with the Academic Editor, we have decided to forward with a decision based on the feedback we have on hand, in an effort to limit further delays to your manuscript. Please note that we will forward you the third review if it is sent to us belatedly.

In light of the reviews, which you will find at the end of this email, we would like to invite you to revise the work to thoroughly address the reviewers' reports. As you will see below, the reviewers agree that your study is generally well done and interesting, but they have each highlighted areas that should be strengthened further in a revision.

Given the extent of revision needed, we cannot make a decision about publication until we have seen the revised manuscript and your response to the reviewers' comments. Your revised manuscript is likely to be sent for further evaluation by all or a subset of the reviewers.

**IMPORTANT - SUBMITTING YOUR REVISION**

*Re-submission Checklist*

*Published Peer Review*

*PLOS Data Policy*

*Blot and Gel Data Policy*

Sincerely,

Luke

Lucas Smith, Ph.D.

Senior Editor

PLOS Biology

lsmith@plos.org

REVIEWS:

Reviewer #1: This is an ambitious and well-executed study combining high-density EEG with a large number of serial awakenings to investigate how dream phenomenology shapes subjective sleep depth. Michalak and colleagues show that immersive NREM dreams counteract the association between cortical activation and shallower perceived sleep, and propose that immersive dreaming helps sustain the felt depth of sleep across the night, even as physiological sleep pressure wanes.

The study potentially makes a strong conceptual contribution to sleep and consciousness research. However, several points require clarification and refinement. Below, I outline major and minor comments to be addressed prior to publication.

Major comments

1. When including the "experience" (exp.) variable in the models, only the interaction with brain activity is reported. It would be important to also report the main effect of experience itself on subjective sleep depth and sleepiness.

2. It would strengthen the modeling approach to include indices of model fit (e.g., AIC, BIC, likelihood-ratio tests) to quantify the variance explained by adding experience or report type variables. This would help assess their explanatory power beyond EEG measures.

3. Report type was modeled as categorical, which is reasonable, but the categories arguably represent gradations of consciousness that are not equidistant. Did the authors also test an ordinal or continuous representation of consciousness? Even if not ideal, discussing why the categorical approach was chosen would clarify the rationale.

4. The construct of "sleep depth" is ambiguous. Here it is operationalized subjectively, but it is often used physiologically (N2/N3, SWA). Since the dataset is restricted to N2, the generalization to "NREM sleep" more broadly should be tempered, ideally already in the title and abstract.

5. The conclusion that immersive dreams "sustain" perceived sleep depth as sleep pressure wanes is compelling but causal framing is too strong. The data are correlational, and immersive dreams might also preferentially arise in physiologically stable deep states. Please temper this language and acknowledge possible bidirectionality.

6. The neural analyses focus mainly on delta, gamma, and their ratio. While justified, this is a relatively coarse view of NREM dynamics. Given the established role of spindles and slow oscillations in sleep depth and mentation, it would be valuable to at least discuss why these features were not included.

7. The category of "conscious experience with sense of presence" (CESP) is intriguing but remains conceptually vague and close to simple CEWR. More justification of how this construct was operationalized and why it is distinct from light dream mentation would be useful.

Minor comments

1. The two experiments differ substantially, with experiment #2 including stimulation. Although "experiment" is modeled, it would be useful to know whether it had a significant effect on outcomes.

2. The "Fig." column in Table 1 does not always correspond to the figure numbering in the manuscript. Please check for consistency.

3. It was not immediately clear that only significant betas are reported in the main text. An explicit clarification would avoid confusion.

4. Awakenings were performed at least 15 minutes after REM offset. Could residual REM dreaming beyond this cutoff still influence reports? Please discuss this possibility.

5. Frequent awakenings may themselves alter dream content or subjective depth perception across the night. While partially controlled statistically, a brief acknowledgment would be appropriate.

6. The finding that UNC is rated as deep but associated with the highest sleepiness deserves more discussion. Why would unconscious sleep feel deep yet leave participants more sleepy than immersive dreams?

Reviewer #2: This is a carefully designed and timely study that combines high-density EEG with a serial awakening paradigm to probe how subjective sleep depth is shaped by both neural and experiential factors. The central and intriguing finding is that perceived depth is not only associated with cortical deactivation, as indexed by spectral EEG measures, but is also strongly modulated by the quality of ongoing mental activity. In particular, immersive dream experiences sustain or even enhance the feeling of deep sleep, counteracting the expected decline in subjective depth as homeostatic sleep pressure dissipates. This challenges the traditional view that perceived "deep sleep" is defined solely by reduced cortical activity and unconsciousness.

From a conceptual perspective, the work opens an interesting discussion: sleep depth may not only be the passive reflection of cortical silence but can also emerge from active brain processes such as immersive dreaming. Such experiences can create a profound sense of detachment and immersion that subjectively "feels deep," even though physiologically they are accompanied by more wake-like activation. This decoupling of cortical state and subjective depth represents a novel contribution and highlights the importance of integrating phenomenological accounts of dreaming with electrophysiological models of sleep.

Overall, the study provides strong data and a valuable perspective on the multidimensional nature of sleep depth. It will be of broad interest to sleep and consciousness researchers, and it invites further work on how experiential richness, not just deactivation, shapes the felt quality of sleep.

On the critical side I believe that the overall messages of the paper are rather simple and for this the paper is overly long and partly highly redundant. I would strongly suggest shortening the message and streamline the main story. The 2nd most critical point for me is that only light N2 sleep awakenings were performed yet strong statements are made about for example REM sleep or immersive dreaming episodes. This needs to be elaborated on.

Major/Minor feedback:

Abstract:

Can the authors please clarify what is meant by that sentence? "Indeed, perceived sleep depth was lowest during states with a mere sense of presence and highest during immersive dreaming or deep unconsciousness.". I am not sure if "mere sense of presence" is clear at that point to the reader and it means physiologically?

I also wonder if one could make that final statement in the abstract clearer: "These results challenge the view that deep sleep stems solely from reduced brain activity and suggest that immersive dreaming sustains perceived sleep depth as homeostatic pressure wanes." I assume the point here is that "deep sleep" as discussed in the public (but not "deep NREM sleep", what scientists automatically will understand reading that sentence) is not merely a deactivated brain but also could stem from a feeling of high sleep depth due to a "close loop" such as in immersive dreaming right? I am picky here as I believe that the abstract should already be crystal clear on its own without the main text.

Methodologically I do not understand the rationale to perform the serial awakening paradigm "specifically targeting N2 sleep" as a strength here would be to exactly probe perceived sleep depth by awakening in various sleep stages including REM and deep N3 sleep?? Please clarify.

Spell out what you mean with a "lower high-to-low frequency power ration" to be 100% clear please. (unfortunately no pages and line numbering is present). In my document this is page 12, last sentence.

The authors state the interesting finding that across the night "perceived sleep depth increased" also objective measures of sleep pressure decreased which is a crucial finding for their main statement. I here wonder if the authors can compare the duration of REM episodes in that 2nd half of the night, or the successive minutes spent in REM per individual and whether this also predicts how deep people perceive their sleep. One could speculate that more minutes spent in REM already in the first half may lead to less intense dreaming (in the 2nd half) and less perceived sleep depth. That would allow to further understand that proposed association of the authors.

Figure 1 is interesting but also needs to state that you only use awakenings out of N2 sleep (if I understand correctly). Do the authors argue that due to the fact that REM gets more in the 2nd half of the night N2 awakenings in the vicinity of REM reflect that state? This is by far my biggest criticism until this point as I find it difficult to argue about immersive dreaming (and associating with REM being cortically not that deactivated) but at the same time not even awakening from REM… In that case I would at least like to see the distribution of how many minutes away where the N2 awakening from the REM episodes in the quarters of the night and read the justification early in text that N2 awakenings close to deep N3 sleep or REM sleep supposedly are biased by content or physiological activations there. Alternatively the authors could analyse how many of their 2min analysis windows actually hit REM or N3 and if these reports were distinct in sleep depth quality as compared to such reaching into N2 also before.

Until the section where I read "Results" as headline I already quite some results. Can this be better separated from the introduction?

Reading the % of conscious experiences, no conscious experiences and CE without recall of content I wonder why they do not sum up to 100%. Please clarify here.

Figure 3 is a bit too complex in my view. E.g. do you really need the scatter plots,… Consider to change and to add what the reader should focus on in these graphs.

What does the sentence "While both dreaming and dreamless sleep are perceived as deeper than minimal experiences, conscious awareness, regardless of its complexity, appears to mitigate feelings of sleepiness upon awakening." mean? I thought with "dreaming" you refer to "conscious awareness" ? Clarify.

Rather than using this unnecessary labeling in Figure 3 "Color coding: CE = green; rich CEWR = dark orange; simple CEWR = light orange; CESP = light purple; UNC = dark purple." (it can be inferred from the bar graphs to the right anyways) the reader would benefit to again have the written out descriptions of these states here.

Next the authors talk about "phenomenological features of conscious experiences" but do not mention that this is a PCA performed on the text scripts of written dreams right? Please clarify on what exactly that PCA is performed on.

Reading this interesting results one of course immediately question oneself that the physiological correlate difference may be here. Please add whether eg more high frequencies in frontal areas may explain the negative association of "reflective thought" and perceived sleep depth. (p.18, 2nd paragraph; "Statistical analyses showed that perceptual immersion was positively associated with subjective sleep depth (β = 0.277, CI = [0.208, 0.346], p < 0.00001), whereas reflective thought was negatively associated with sleep depth (β = -0.195, CI = [-0.287, -0.102], p < 0.00001).")

The authors repeatedly talk about EEG activity and spectra but should there also add if this is always a grand-average or if specific topographies are here in the focus. Eg. For the association of Gamma and subjective sleepiness one wonders if this holds true for the whole brain or only certain parts of the brain.

Discussion

Also here I think it is crucial to clearly distinguish subjectively perceived "deep sleep" and "deep NREM sleep" as scientists may read sentences as the following: "Contrary to the longstanding view that deep sleep…"

Please rephrase for better readability and clarifying your main point: "wake-like high-frequency activity... is perceived as less disruptive when accompanied by conscious experience."

Methodologically I wonder how accurate the U-Sleep automatic algorithm is for sleep staging and whether a subset was also manually scored for validation?

Please clarify. Awakenings were not dependent on that algorithm but performed online by well trained personnel?

Awakening criteria or instructions for experiment #1 and #2 were completely identical?

---

## [Decision Letter · Decision Letter 2]

30 Jan 2026

Dear Dr Bernardi,

Thank you for your patience while we considered your revised manuscript "Immersive NREM2 dreaming preserves subjective sleep depth against declining sleep pressure" for publication as a Research Article at PLOS Biology. This revised version of your manuscript has been evaluated by the PLOS Biology editors, the Academic Editor and the original reviewers, who are largely satisfied with the revision.

Based on the reviews, we are likely to accept this manuscript for publication, provided you satisfactorily address the remaining points raised by the reviewer 2. We think reviewer 2 has raised some important points, but think these can be largely addressed by providing additional clarifications and tweaks to the presentation (rather than requiring new analyses).

**IMPORTANT: As you address reviewer 2's last requests, we also ask that you address the following data and other policy-related requests.

1) ABSTRACT: Please note that per journal policy, the model system/species studied should be clearly stated in the abstract of your manuscript. Please update the abstract to explicitly state that human subjects were studied here.

2) ETHICS STATEMENT: Please update your ethics statement, in your methods section, to indicate whether the study was conducted according to the principles expressed in the Declaration of Helsinki.

3) DATA: Thank you for providing your underlying data and code on zenodo. Please update each figure legend with a brief sentence pointing readers to this dataset.

We expect to receive your revised manuscript within two weeks, but we would be happy to provide more time, if needed. Please email plosbiology@plos.org to request an extension.

*Published Peer Review History*

*Press*

Sincerely,

Luke

Lucas Smith, Ph.D.

Senior Editor

lsmith@plos.org

PLOS Biology

Reviewer remarks:

Reviewer #1: The authors addressed all my concerns. I have nothing more to add.

Reviewer #2, Manuel Schabus: This is a very good and detailed revision. Yet I question the argument that NREM-2 awakenings were chosen as this is a more homogeneous state… I believe it is also simply methodological easier to conduct the experiment that way and not fully it is still not fully understandable to me given that N3 and specifically REM dreams are discussed in the literature. But it with no doubt is a very appropriate and sensible revision and a very interesting paper for a broad audience. It could be discussed that the argument that "repeated interruptions of N3 early in the night could plausibly affect perceived sleep depth in later N2" is perhaps even a valid argument for "N2 only" awakenings.

In the same direction I see the answer to the issue: "the established role of spindles and slow oscillations in sleep depth and mentation, it would be valuable to at least discuss why these features were not included." Also here the authors come up with an elegant answer that they just computed frequency band associations hypothesis-driven like previous papers (Siclari et al., 2017); yet that way scientists never could find new associations and intriguing results. I would like to see at least a sentence in the discussion saying that these are microevents that definitely would be worth analyzing in future studies.

Lastly I suggest that Figures like Figure 5 also have the end of scales in the axis. E.g. here "high" and "low" subjective sleep depth (panel A), etc. next to the y-axis (left and right) to be more intuitive to understand.

---

## [Editor Report · Decision Letter 3]

18 Feb 2026

Dear Dr Bernardi,

Thank you for the submission of your revised Research Article "Immersive NREM2 dreaming preserves subjective sleep depth against declining sleep pressure" for publication in PLOS Biology and thank you for addressing the last reviewer and editorial requests in this revision. On behalf of my colleagues and the Academic Editor, Matthew P. Walker, I am pleased to say that we can in principle accept your manuscript for publication, provided you address any remaining formatting and reporting issues. These will be detailed in an email you should receive within 2-3 business days from our colleagues in the journal operations team; no action is required from you until then. Please note that we will not be able to formally accept your manuscript and schedule it for publication until you have completed any requested changes.

PRESS

Sincerely,

Luke

Lucas Smith, Ph.D.

Senior Editor

PLOS Biology

lsmith@plos.org